# A Nexus between Genetic and Non-Genetic Mechanisms Guides KRAS Inhibitor Resistance in Lung Cancer

**DOI:** 10.3390/biom13111587

**Published:** 2023-10-28

**Authors:** Prakash Kulkarni, Atish Mohanty, Sravani Ramisetty, Herbert Duvivier, Ajaz Khan, Sagun Shrestha, Tingting Tan, Amartej Merla, Michelle El-Hajjaoui, Jyoti Malhotra, Sharad Singhal, Ravi Salgia

**Affiliations:** 1Department of Medical Oncology and Experimental Therapeutics, City of Hope National Medical Center, 1500 Duarte Rd., Duarte, CA 91010, USA; amohanty@coh.org (A.M.); sramisetty@coh.org (S.R.); jymalhotra@coh.org (J.M.); ssinghal@coh.org (S.S.); 2Department of Systems Biology, City of Hope National Medical Center, 1500 Duarte Rd., Duarte, CA 91010, USA; 3Department of Medical Oncology, City of Hope Atlanta, 600 Celebrate Life Parkway, Newnan, GA 30265, USA; herbert.duvivier@ctca-hope.com; 4Department of Medical Oncology, City of Hope Chicago, 2520 Elisha Avenue, Zion, IL 60099, USA; ajaz.khan@ctca-hope.com; 5Department of Medical Oncology, City of Hope Phoenix, 14200 West Celebrate Life Way, Goodyear, AZ 85338, USA; sagun.shrestha@ctca-hope.com; 6Department of Medical Oncology, City of Hope National Medical Center, Newport Beach Fashion Island, Duarte, CA 92660, USA; titan@coh.org; 7Department of Medical Oncology, City of Hope, Lancaster, CA 93534, USA; amerla@coh.org; 8Department of Medical Oncology, City of Hope Medical Center, West Covina, CA 91790, USA; melhajjaoui@coh.org

**Keywords:** KRAS mutation, drug resistance, non-genetic mechanisms, sotorasib, adagrasib

## Abstract

Several studies in the last few years have determined that, in contrast to the prevailing dogma that drug resistance is simply due to Darwinian evolution—the selection of mutant clones in response to drug treatment—non-genetic changes can also lead to drug resistance whereby tolerant, reversible phenotypes are eventually relinquished by resistant, irreversible phenotypes. Here, using KRAS as a paradigm, we illustrate how this nexus between genetic and non-genetic mechanisms enables cancer cells to evade the harmful effects of drug treatment. We discuss how the conformational dynamics of the KRAS molecule, that includes intrinsically disordered regions, is influenced by the binding of the targeted therapies contributing to conformational noise and how this noise impacts the interaction of KRAS with partner proteins to rewire the protein interaction network. Thus, in response to drug treatment, reversible drug-tolerant phenotypes emerge via non-genetic mechanisms that eventually enable the emergence of irreversible resistant clones via genetic mutations. Furthermore, we also discuss the recent data demonstrating how combination therapy can help alleviate KRAS drug resistance in lung cancer, and how new treatment strategies based on evolutionary principles may help minimize or even preclude the emergence of drug resistance.

## 1. Introduction

For over a century, starting with Theodore Boveri’s ground-breaking observations in the early 1900s [1,2], cancer has been thought to be a genetic disease [3,4]. The ensuing period provided a reductionist perspective and further helped to firmly establish that cancer is a complex, heterogeneous disease driven by genetic mutations. In fact, it was recently suggested that cancer may be defined as “a disease of uncontrolled proliferation by transformed cells subject to evolution by natural selection” [5]. The latter part of the definition, namely “subject to evolution by natural selection”, underscores how the inexorable interaction between the genotype and the environment gives rise to a certain phenotype which, in this case, happens to be cancer. It is therefore not surprising that mutations in oncogenes that regulate cell growth and division are often seen in many types of cancers. However, this correlation is rather tissue-specific. For example, the Kirsten rat sarcoma 2 viral oncogene homolog (KRAS) is an oncoprotein that is mutated in a majority of cancer types, but the mutations occur either at different sites in the polypeptide sequence or at the same residue but with different substitutions. For instance, KRAS G12 mutations are quite common in many cancers; however, while G12D is predominant in pancreatic cancer, G12C is the most frequent mutant form in non-small cell lung cancer (NSCLC) [6] (Figure 1). Why?

The answer to this important question, according to the prevailing wisdom, is that the transformed cell is trying to adapt itself to the rugged fitness landscape that is molded by the stressful environment that causes cancer. In doing so, only those mutations that increase the cell’s fitness are favored over those that do not significantly increase its fitness. And since the landscape is different in various tissues and organs, different mutant oncogenes, or different mutations in the same oncogene (as seen in the case of KRAS), are selected to maximize the chances of a cancer cell adapting itself.

An even more fundamental question with regard to the above example is, why is KRAS so frequently mutated? Granted, KRAS is a component in the signaling pathway that regulates cell proliferation, and hence a mutation that renders KRAS constitutively active would be an obvious choice. But some of the other players in the same pathway, such as the upstream molecule EGFR or the downstream molecules BRAF, MEK and ERK, are not the preferred candidates for mutation and constitutive activation. In fact, one rarely encounters mutations in many of these genes [7]. Why?

On the other hand, heretical to the prevailing wisdom, cancer cells burdened with the canonical causative mutations can be reprogrammed into pluripotent stem cells, and cancer can result even in the absence of mutations in a so-called ‘driver’ oncogene. For example, c-Myc, which is mutated in several types of cancer, can also cause cancer by simply being overexpressed in its wildtype form (sans mutations) [8], calling into question the genetic basis of the disease. Furthermore, dialing down Myc expression can override the transformed phenotype to revert it to a non-transformed state, alluding to a non-genetic underpinning.

The idea that phenotypic plasticity and non-genetic mechanisms are also important in disease etiology, progression, metastasis, heterogeneity and drug resistance was suggested a decade ago [9] but these pathways are only now being increasingly recognized as being just as important as the genetic mechanism [10]. Indeed, growing evidence indicates that phenotypic plasticity and non-genetic mechanisms are not only critical, but are also leveraged as a bet-hedging strategy in cancer [11,12,13]. Furthermore, the realization that a ‘persister’ population may also contribute to drug resistance in cancer [14,15,16,17,18] further underscores the importance of both genetic and non-genetic mechanisms of drug resistance in cancer.

### The KRAS Inhibitor Resistance Paradigm

Despite decades of research and a detailed molecular understanding of the pathways that KRAS activates, until recently, no small molecule inhibitors of KRAS were available and, in fact, KRAS was believed to be an ‘undruggable’ target [6]. However, in just the past few years, several novel therapeutics have transformed the landscape of treatment for patients with mutant KRAS [19,20,21]. Small-molecule inhibitors that are either specific to G12C, such as sotorasib (AMG510) and adagrasib (MRTX849) [22,23,24], or that are non-G12C-specific (pan-KRAS) inhibitors, have been developed [25]. While pan-KRAS inhibitors are currently being evaluated, both sotorasib and adagrasib have been approved by the FDA for the treatment of patients with KRAS G12C-mutated advanced NSCLC with progression in prior chemotherapy and/or checkpoint inhibitors based on results from the CodeBreak trial and KRYSTAL-1, respectively [26,27]. However, accumulating evidence from preclinical and clinical studies has shown that, these KRAS G12C targeted therapeutics, as single agents, inevitably result in drug resistance, a persistent problem associated with targeted therapies, emphasizing the need to explore combination treatments with other therapeutic agents.

A number of clinical trials are investigating the benefit from adding other modalities, such as immunotherapy, to KRAS inhibitors to improve clinical efficacy (Table 1). But these studies so far have not shown significant promise. Jänne et al. recently reported results from the KRYSTAL-1/KRYSTAL-7 trials at the ESMO Immuno-Oncology Congress in December [28]. These trials reported results from 75 patients with KRAS G12C-mutated advanced NSCLC who were treated with adagrasib and a PD-1 inhibitor, pembrolizumab. While the combination was well tolerated, response rates in the two trials ranged from 49 to 57% only. Similarly, the combination of sotorasib with the MEK inhibitor, trametinib, has also not shown promising results for KRAS G12C-mutated NSCLC. The response rates were only 20% in patients who were sotorasib-naïve and 0% in patients with KRASi resistance, as reported in the CodeBreak 101 trial [29].

Several studies have helped to illustrate the issues associated with drug resistance caused by KRAS inhibitors [31,32,33,34,35,36]. While some studies have identified secondary mutations in the KRAS molecule per se, highlighting the role of genetic mechanisms, other studies have alluded to non-genetic mechanisms contributing to drug resistance. For example, in one study, 38 patients with KRASG12C-mutated solid tumors, that included 27 patients with non-small-cell lung cancer, 10 with colorectal cancer, and 1 with appendiceal cancer were treated with adagrasib [37]. The authors found that genetic alterations accounted for resistance in many patients with KRAS G12C-mutated colon cancer. However, this was not the case for patients with KRAS G12C-mutated NSCLC. These tumors became resistant in the absence of apparent genetic mutations in ~50% of these patients. Furthermore, the same study also showed that myriad on-target and off-target mechanisms can confer resistance to KRAS G12C inhibitors [37].

For the past several years, we have been investigating the non-genetic mechanisms underlying drug resistance in lung cancer, including group behavior and persistence [9,38,39,40,41,42,43]. These studies have provided an elegant conceptual framework reconciling the genetic/non-genetic duality of cancer drug resistance and have helped to develop an understanding of how drug-resistant cancer cells can also arise from sensitive cells via an intermediate drug-tolerant state. They also showed how cancer cells can switch their phenotypes to evade drug effects [42]. Thus, in contrast to the current notion that drug resistance is simply due to Darwinian evolution (i.e., the selection of mutant clones in response to drug treatment), our data, as well as those from several others [38,43,44,45,46,47], suggest that in response to drug treatment, both non-genetic and genetic changes lead to drug resistance. In the case of the former, tolerant phenotypes are eventually relinquished, but they can temporarily guard the tumor against extinction while enabling the emergence of more permanent resistance mechanisms. Here, using KRAS as a paradigm, we illustrate how this nexus between genetic and non-genetic mechanisms enables cancer cells to evade the harmful effects of drug treatment. Furthermore, we also discuss the recent data demonstrating how the FDA-approved proteasome inhibitor carfilzomib (CFZ), in combination with sotorasib, can alleviate drug resistance in lung cancer.

## 2. Phenotypic Plasticity and Drug Sensitivity

Drug resistance, a term that is now vernacular, is generally thought to occur via genetic alterations that are irreversible [48,49,50,51,52,53,54], and no explicit delineation of inherent or acquired resistance is implied. Inherent resistance refers to preexisting genetic changes that are irreversible, arise by chance and prevent cancer cells from responding to therapy altogether [48]. In contrast, acquired resistance that arises after the fact and is also eventually irreversible, can be due to a mutation or can be the result of a reversible tolerant state. This reversible phenotype allows the cancer cells to revert to their original state and repopulate, leading to tumor proliferation in the absence of the drug. In clinical terms, the tolerant state is analogous to stable diseases in patients. However, the prolonged exposure of tolerant cells to the drug can lead to mutations in tolerant cells, leading to the development of irreversible resistance. While the genetic basis of drug resistance is well recognized, the non-genetic mechanisms leading to a tolerant state, that eventually leads to acquired resistance, are only now beginning to be appreciated.

To understand how the disease is initiated and how cancer cells evade drug effects at a more fundamental level, we previously put forth the conformational noise hypothesis, which explains how rewiring of the protein interaction network (PIN) can lead to disease [9]. Here, noise is the random variability in quantities arising in biological systems, such as the variability in the number of transcription factors between two different cells in the population, even when the cells are isogenic. We stressed the fact that, like all other life forms, cancer cells are complex adaptive systems. They are dissipative structures that operate far from thermodynamic equilibrium and exhibit emergent properties. Their extraordinary ability to self-organize and thus adapt themselves to the fitness landscape is driven by their protein PIN, which serves as the main conduit for information flow in the system. Cellular PINs follow a scale-free architecture wherein intrinsically disordered proteins (IDPs) typically occupy hub positions. IDPs are proteins (or regions within ordered proteins) that lack a rigid structure and exist as conformational ensembles whose preferences are dictated by covalent modifications such as the phosphorylation of the amino acid side chains. Therefore, we postulated that IDPs can contribute to noise in the system, which we referred to as conformational noise to distinguish it from the well-recognized transcriptional noise. Furthermore, by virtue of their innate ability to interact with multiple partners, especially when overexpressed, IDPs can engage in promiscuous interactions to rewire the PIN. Thus, we posited that noise-driven PIN rewiring actuates phenotypic switching [9].

IDP overexpression occurs in response to stress, but upon stress withdrawal, expression returns to normal levels and cells can switch back to their earlier phenotype. However, chronic stress causes PIN frustration and mutations in certain hotspots, including key IDPs, which can relieve the PIN frustration. Remarkably, ~80% of cancer-associated proteins (including most oncoproteins), ~90% of cellular transcription factors (including the well-known ‘Yamanaka factors’ involved in cellular reprogramming), a large fraction of proteins involved in signal transduction and the majority of stress response proteins are IDPs, lending further credence to the conformational noise hypothesis [9,10].

## 3. KRAS Conformational Dynamics and Its Sensitivity to Inhibitors

KRAS, a small GTPase of 189 amino acids, is a constituent of the RAS/MAPK pathway. Although billed as an ordered protein, with extensive X-ray crystallography data to support this claim, KRAS has significant intrinsically disordered regions (IDRs) interdigitated between its highly ordered regions, and a C-terminal tail (amino acids 170–189), referred to as the hypervariable region, that is entirely intrinsically disordered (Figure 2). Binding to GTP activates KRAS, and its hydrolysis to GDP turns it OFF. This ON/OFF mechanism enables the protein to transduce signals that instruct the cell regarding whether to proliferate or differentiate. Mutations in the KRAS oncoprotein are the most common gain-of-function alteration in many cancer types, including lung cancer, where they account for ~30% of lung adenocarcinomas in the western world. Codon12 in KRAS is a hotspot for oncogenic mutations such as G12A, G12C, G12D, G12R, G12S, and G12V, disturbing the active fold of the protein. Functionally, these point mutations impair its GTPase activity and thus render the oncoprotein constitutively active. Furthermore, the different conformations imposed by distinct amino-acid substitutions in the KRAS molecule are thought to cause the altered activation of downstream signaling pathways, such as Raf/MEK/ERK, thus leading to distinct clinical outcomes [55].

Physics-based computational studies reveal that the KRAS molecule, in its native (unbound) apo form, exists as a conformational ensemble that has considerable flexibility due to the presence of the IDRs [56,57]. All-atom molecular dynamics (MD) simulations showed that sotorasib does not entrap KRAS in a single conformation, as one would expect based on the crystal structure, but rather in an ensemble of conformations [56], as is typical of an IDP. In contrast, upon binding to GDP/GTP, the ensemble becomes significantly structured; however, the various point mutations bias the conformational preferences of the holoenzyme. Therefore, although, functionally, these mutations impair KRAS’s GTPase activity (lock it in the bound state) and render the oncoprotein constitutively active, they impinge on the selection of the partners with which KRAS interacts and thus impact downstream signaling and potentially affect clinical outcomes. One potential mechanism underlying such differential partner interactions could be the significant differences in the binding affinities in protein–protein interactions (PPI). A recent PPI study, using three homologous protease-inhibitor PPIs that spanned nine orders of magnitude in binding affinity, found that quantitative binding landscapes consisting of ΔΔGbind values for the three PPIs were vastly different depending on specific mutations [58]. Consistent with this hypothesis, in lung cancer, tumors with KRAS G12C exhibit greater ERK1/2 phosphorylation than those with KRAS G12D [59]. Indeed, KRAS G12C tumors are significantly more sensitive to MEK inhibitors than are KRAS G12D tumors, and inhibiting MEK in KRAS G12C mice significantly increased chemotherapeutic efficacy and progression-free survival (PFS) [59]. Furthermore, after initially responding well, ERK-mediated feedback inhibition of the vertical RTKs/SHP2 pathway is lifted, which induces new synthesis and the activation of KRAS G12C and facilitates KRAS promiscuity. Together, these observations indicate that different amino acid substitutions in oncogenic KRAS lead to heterogeneity in the biological behaviors of the mutant protein and that KRAS signaling can crosstalk with alternative signaling pathways to contribute to drug resistance [55].

Additional computational studies employing long-time MD simulations (75 s in total) have examined the structural and energetic features of KRAS G12C and its four drug-resistant variants resistant to inhibitors [60]. Strikingly, the combined binding free energy calculation and the protein–ligand interaction fingerprint revealed that secondary mutations caused KRAS to produce different degrees of resistance to AMG510 and MRTX849. Markov State Models and 2D free energy landscape analysis showed a difference in the conformational changes seen in mutated KRAS in the absence/presence of inhibitors [60]. Further, a comparative analysis of these systems showed that there were differences in their allosteric signal pathways, highlighting the strong possibility that different conformations imposed by distinct amino acid substitutions in the KRAS molecule alter PPIs.

## 4. The Focal Adhesion Complex and Drug Resistance

Consistent with the role of IDPs in actuating non-genetic mechanisms that eventually lead to irreversible drug-resistance phenotypes [9,61,62], we previously showed that the IDPs associated with the focal adhesion complex, namely integrin beta 4 (ITGB4) and paxillin (PXN), can induce cisplatin resistance in KRAS-mutant NSCLC via non-genetic mechanisms [40,63]. The co-expression of these proteins correlated with poor patient survival, and the perturbation of their signaling using CFZ led to cell growth inhibition and sensitization to cisplatin [40]. However, the contribution of these two proteins to acquiring tolerance or resistance to sotorasib has remained poorly understood.

Using isogenic cell lines, we recently demonstrated that acquired resistance to sotorasib was correlated with an increased expression of integrin β4 (ITGB4) [30]. While knocking down ITGB4 in tolerant cells improved sotorasib sensitivity, its overexpression enhanced tolerance to sotorasib by activating the AKT-mTOR bypass pathway. On the other hand, chronic sotorasib treatment induced WNT expression and activated the WNT/β-catenin signaling pathway. However, silencing the expression of both genes significantly improved sotorasib sensitivity, not only in drug-tolerant cells, but also in cells with acquired resistance, as well as in those cells that were inherently resistant to sotorasib. Furthermore, the pharmacological downregulation of ITGB4 and β-catenin expression via CFZ showed synergism with sotorasib in alleviating drug resistance. Surprisingly, in inherently sotorasib-resistant cells, adagrasib phenocopied the synergistic effect of the sotorasib/CFZ combination by suppressing KRAS activity and inhibiting cell cycle progression. Together, these findings uncovered the novel non-genetic mechanisms underlying sotorasib resistance and identified a potential therapeutic strategy to overcome resistance.

## 5. Conclusions

While mutations are the source of new variation, natural selection does not create new traits; it only changes the proportion of variation that is already present in the population. However, selection is not a ‘passive’ process, either. The unmasking of cryptic variations, as elegantly demonstrated by Conrad Hal Waddington (1957) [64], or the introduction of novel mutations in response to specific stress that an organism, in this case, a cancer cell, experiences, underscore the active epigenetic (in Waddington’s terminology) interaction between the environment and the genotype. Therefore, it is this iterative two-step interaction between these processes that leads to the evolution of novel adaptive features and not just random mutations dictating the ‘fitness’ of the cancer cell. In addition, non-genetic mechanisms, such as PIN rewiring and epigenetic modifications (e.g., DNA methylation changes), in response to environmental perturbations, acting independently or together with the genetic mechanisms, contribute to adaptive evolution (Figure 3). Indeed, it is now becoming increasingly evident that single cells can ‘anticipate’, ‘learn’ and ‘compute’ to ‘make’ appropriate decisions [65,66,67,68,69], underscoring the fact that cancer cells are adept at adapting themselves. Therefore, the PIN-rewiring heuristic facilitated by IDPs such as KRAS in response to an environmental perturbation, through which the cancer cell ‘learns’ about or senses its environment, reflects its attempts to negotiate the fitness landscape. Alternatively, like Denis Noble states in his famous essay, ‘Central Dogma or Central Debate’ [70], a purely genetically determined mechanism would be akin to how, ‘The unidirectionality of sequence information transfer from DNA to proteins no more determines life than the QWERTY keyboard determines what I wrote in this article.’

As mentioned above, yet another factor that has huge implications for how cancer cells evade the harmful effects of a drug, and hence improve their fitness, is group behavior. Resistance results from a complex interplay between groups of cells within a heterogeneous population and the surrounding tumor microenvironment. Therefore, phenotypic plasticity, intercellular communication and adaptive stress response act in concert to ensure that, in the absence of a priori genetic events such as mutations, intermediate reversible phenotypes can withstand drug effects until permanent, resistant clones can emerge. Thus, understanding the role of group behavior is paramount to developing effective treatment strategies while minimizing, or at least delaying, the emergence of bona fide resistance. In light of this new thinking, perhaps it may be prudent to address the emergence of drug resistance in the clinical setting by considering alternate therapeutic strategies based on evolutionary game theory, such as ‘adaptive’ therapy [71]. Current treatment protocols have typical leader–follower (or “Stackelberg”) dynamics; the “leader”, in this case, the oncologist, plays first by administering the therapy, and the “followers”, in this case, the cancer cells, then respond and adapt to the therapy. Thus, by repeatedly administering the same drug or drug combinations until the disease progresses, the oncologist, albeit inadvertently, enacts a fixed strategy, while the oncologist’s opponents (the cancer cells) continuously evolve strategies to successfully adapt to the drug treatment [72].

## Figures and Tables

**Figure 1 biomolecules-13-01587-f001:**
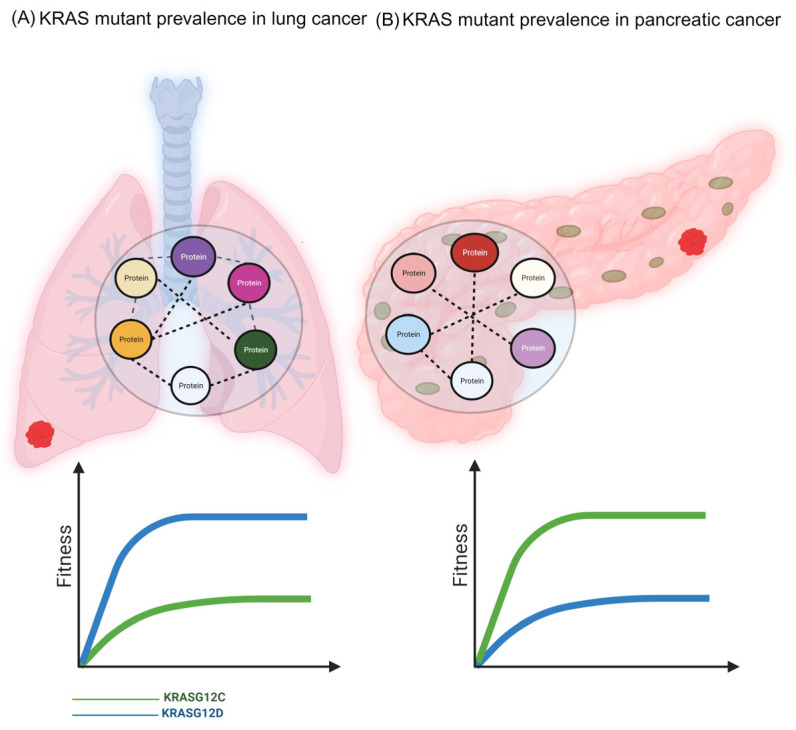
KRAS codon 12 mutational signature in lung and pancreatic cancer landscape. (**A**) KRAS G12C mutation is more prevalent in non-small cell lung cancer (~40%) compared to the KRAS G12D mutation. (**B**) Conversely, in pancreatic cancer, KRASG12D mutation is the most frequently observed mutation. Cancer cells with the respective mutations are favored for their fitness in the fitness landscapes prevailing in the lung and pancreas, respectively. Figure 1 was made using BioRender.com (1 September 2023).

**Figure 2 biomolecules-13-01587-f002:**
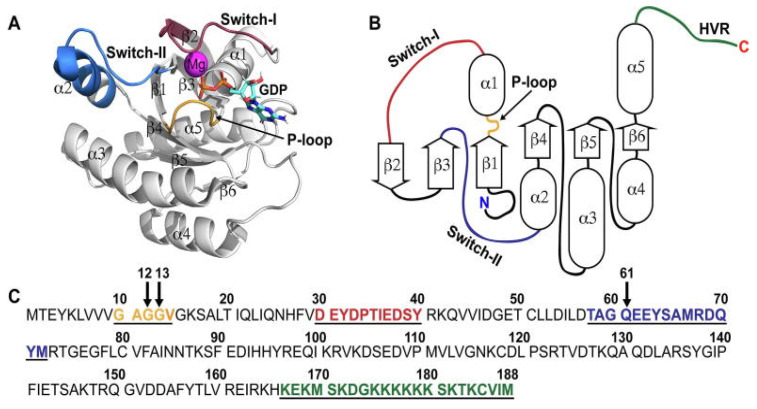
Structure and sequence of KRAS4B. (**A**) Crystal structure of wild-type (WT) KRAS with GDP bound (PDB ID: 4obe). The C-terminal HVR is not present in the structure. (**B**) 2D depiction of the secondary structure of KRAS. Disordered regions are indicated by lines. (**C**) Sequence of KRAS4B. The most common mutation hotspots are indicated by arrows. Selected structural regions in all (**A**–**C**) are highlighted with the following color scheme: P-loop (residues 10–14), orange; switch-I (residues 30–40), red; switch-II (residues 58–72), blue; HVR (residues 167–188), green. Adapted from [56].

**Figure 3 biomolecules-13-01587-f003:**
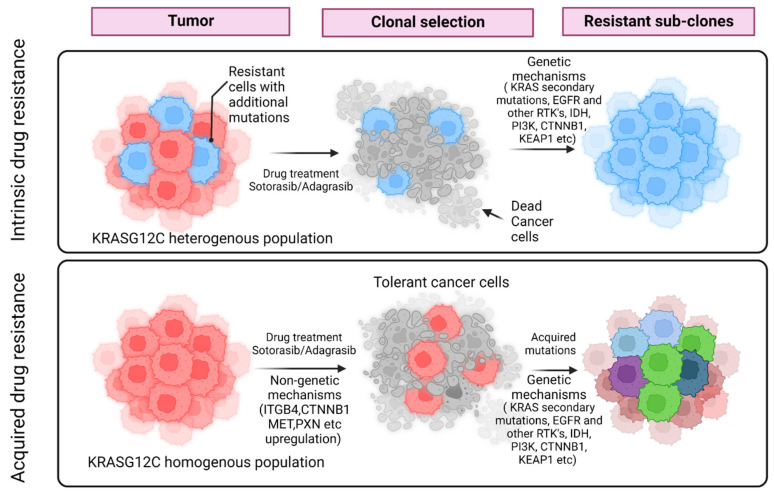
Schematic representation of intrinsic and acquired drug resistance. In the case of intrinsic resistance, tumor cells can become resistant due to mutations that exist a priori. In the case of acquired resistance they can acquire resistance through non-genetic mechanisms initially and then followed by genetic changes. Figure 3 was made using BioRender.com (1 September 2023).

**Table 1 biomolecules-13-01587-t001:** KRAS G12C inhibitors and combination treatments.

Number	Combination	Clinical Study	Reference
1	Adagrasib + immune checkpoint inhibitor pembrolizumab	Yes	[28]
2	Sotorasib + MEK inhibitor trametinib	Yes	[29]
3	Sotorasib + proteasome inhibitor carfilzomib	No	[30]
4	Adagrasib + proteasome inhibitor carfilzomib	No	[30]

## Data Availability

The data presented in this study are available in this article.

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
