# Peer review of "A Nexus between Genetic and Non-Genetic Mechanisms Guides KRAS Inhibitor Resistance in Lung Cancer"

_biomolecules, 2023, doi:10.3390/biom13111587_

Round 1

Reviewer 1 Report

The authors review the interface of genetic and non-genetic determinants of cancer in inducing drug resistance with focus on the role of intrinsically disordered regions of proteins in reshaping protein interactions networks. They focus on KRAS and intergrins to develop the discussion. The idea is interesting and intuitively plausible. My only suggestions is to use in-silico analysis to show by an example how changes in disordered regions of KRAS alter the PIN, as this seems to be a central point in their argument.  

Author Response

Authors response: We thank the reviewer for appreciating our work. We agree that including a Section describing how conformational noise can cause PIN rewiring albeit in silico, would be great. However, we believe this is a lot more nuanced and quite honestly, a topic for another research article rather than this perspective article where the inclusion of original research data is discouraged. Furthermore, we are currently working on wrapping up a manuscript describing how PIN rewiring occurs due to changes in the conformational dynamics of the intrinsically disordered regions of KRAS multiple experimental techniques and mathematical simulations, as a follow up to the KRAS paper we just published in Science Advances (Mohanty et al, "Acquired resistance to KRAS G12C small- molecule inhibitors via genetic/nongenetic mechanisms in lung cancer", 2023, in Press) listed as ref #63 in the current manuscript. Hence, divulging these unpublished results in this perspective article would compromise our second paper on KRAS. We trust the reviewer will understand our predicament.

Reviewer 2 Report

This perspective article tries to use the paradigm where lung cancer is resistant to KRAS inhibitors to hypothesize that cancer cells may actuate a nexus between genetic and non-genetic mechanisms, reconciling and explaining why drug resistance in cancer cells can be caused either by genetic changes due to Darwinian evolution or by non-genetic changes for a transition from reversible to irreversible phenotypes. The perception underlying such a nexus is the conformational noise hypothesis reasoning why and how the intrinsically disordered proteins (IDP) after cancer cells encounter anti-cancer drugs can result in either genetic or non-genetic changes to rewire protein interaction network (PIN), leading to drug resistance in evading the harmful effects of drug treatments. Overall, the perspective is interesting and warrants further investigation. I only have some suggestions for the manuscript which to me is well written.

1. I suggest that the authors concisely illustrate the conformational noise hypothesis in the Abstract to make the "nexus" more understandable.

2. "non-small-cell lung cancer" in Lines 121-122 on page 3 should be replaced by NSCLC and it should be present in the text where NSCLC is to be abbreviated. I suggest that it be abbreviated in the Abstract when the authors describe " ------ can help alleviate KRAS drug resistance in lung cancer ------".

3. In Lines 143-145 on page 4, the authors wrote " ------ how the FDA-approved proteasome inhibitor carfilzomib (CFZ), in combination with sotorasib, can alleviate drug resistance in lung cancer" in Section 1. I suggest that they briefly give a reason why sotorasib is combined with CFZ for one to catch their logic, although I understand that they explain it in Section 4.

4. What is "KRAS G12C51"? Is 51 simply referred to the reference #51?

Author Response

This perspective article tries to use the paradigm where lung cancer is resistant to KRAS inhibitors to hypothesize that cancer cells may actuate a nexus between genetic and non-genetic mechanisms, reconciling and explaining why drug resistance in cancer cells can be caused either by genetic changes due to Darwinian evolution or by non-genetic changes for a transition from reversible to irreversible phenotypes. The perception underlying such a nexus is the conformational noise hypothesis reasoning why and how the intrinsically disordered proteins (IDP) after cancer cells encounter anti-cancer drugs can result in either genetic or non-genetic changes to rewire protein interaction network (PIN), leading to drug resistance in evading the harmful effects of drug treatments. Overall, the perspective is interesting and warrants further investigation. I only have some suggestions for the manuscript which to me is well written.

  1. I suggest that the authors concisely illustrate the conformational noise hypothesis in the Abstract to make the "nexus" more understandable.

Authors response: We thank the reviewer for appreciating our work as well as for the thoughtful and constructive comments. We have now revised the Abstract accordingly to illustrate how the conformational noise hypothesis may help establish the nexus between genetic and non-genetic mechanisms in drug resistance. The changes are highlighted in the revised MS.

  1. "non-small-cell lung cancer" in Lines 121-122 on page 3 should be replaced by NSCLC and it should be present in the text where NSCLC is to be abbreviated. I suggest that it be abbreviated in the Abstract when the authors describe " ------ can help alleviate KRAS drug resistance in lung cancer ------".

Authors response: This has been addressed. The changes are highlighted in the revised MS.

  1. In Lines 143-145 on page 4, the authors wrote " ------ how the FDA-approved proteasome inhibitor carfilzomib (CFZ), in combination with sotorasib, can alleviate drug resistance in lung cancer" in Section 1. I suggest that they briefly give a reason why sotorasib is combined with CFZ for one to catch their logic, although I understand that they explain it in Section 4.

Authors response: We appreciate the reviewer’s suggestion; however, we believe describing the reason to employ the CFZ combination upfront, rather than in Section 4, would be kind of repetitious. Thus, we prefer to leave it as is.

  1. What is "KRAS G12C51"? Is 51 simply referred to the reference #51?

Authors response: That is correct! And this has now been fixed. The changes are highlighted in the revised MS.

Reviewer 3 Report

Genetic alterations are the primary cause of the complicated, diverse disease known as cancer. In various malignancies, mutations in oncogenes that control cell growth and division are frequently observed. Most cancer types have mutations in the Kirsten rat sarcoma 2 viral oncogene homolog (KRAS), either at distinct locations in the polypeptide sequence or at the same position but with various substitutions. While KRAS G12D is more prevalent in pancreatic cancer than G12C in non-small cell lung cancer, KRAS G12 mutations are prevalent in numerous malignancies. According to this perspective, the authors utilized the KRAS gene as a model to show how the interaction of genetic and non-genetic pathways allows cancer cells to resist the adverse effects of drugs. They also explained how novel therapeutic approaches based on evolutionary concepts may help decrease or even prevent the establishment of drug resistance. The article is well written; however, I have a minor concern. 

I suggest the authors provide a table listing combinations of treatments that can lessen KRAS drug resistance in lung cancer. Additionally, if any combination clinical trials for the KRAS inhibitors are available, these should be considered. 

Author Response

Authors response: We appreciate the reviewer’s comments as well as the suggestion to provide a table listing combinations of treatments that can lessen KRAS drug resistance in lung cancer. This table has now been included in the revised manuscript which describes various combination treatments both clinical and preclinical (Table 1). The changes are highlighted in the revised MS.

Round 2

Reviewer 1 Report

The authors have responded to all my comments.